# Inertial Motion Capture-Driven Digital Human for Ergonomic Validation: A Case Study of Core Drilling

**DOI:** 10.3390/s24185962

**Published:** 2024-09-13

**Authors:** Quan Zhao, Tao Lu, Menglun Tao, Siyi Cheng, Guojun Wen

**Affiliations:** 1School of Future Technology, China University of Geosciences, Wuhan 430074, China; zhaoquan@cug.edu.cn (Q.Z.); lutao@cug.edu.cn (T.L.); 2School of Mechanical & Electronic Information, China University of Geosciences, Wuhan 430074, China; 1202220839@cug.edu.cn (M.T.); chengsiyi@cug.edu.cn (S.C.); 3Hubei Intelligent Geological Equipment Engineering Technology Research Center, Wuhan 430074, China

**Keywords:** ergonomics, digital human, motion capture, driller’s cabin, comfortability

## Abstract

In the evolving realm of ergonomics, there is a growing demand for enhanced comfortability, visibility, and accessibility in the operation of engineering machinery. This study introduces an innovative approach to assess the ergonomics of a driller’s cabin by utilizing a digital human. Through the utilization of inertial motion capture sensors, the method enables the operation of a virtual driller animated by real human movements, thereby producing more precise and realistic human–machine interaction data. Additionally, this study develops a simplified model for the human upper limbs, facilitating the calculation of joint forces and torques. An ergonomic analysis platform, encompassing a virtual driller’s cabin and a digital human model, is constructed using Unity 3D. This platform enables the quantitative evaluation of comfortability, visibility, and accessibility. Its versatility extends beyond the current scope, offering substantial support for product development and enhancement.

## 1. Introduction

Ergonomics, through the analysis of human–environment interface interactions, assesses how factors such as the working environment, workstation layout, working posture, and procedures affect human physiology, psychology and productivity [1,2,3]. The objective is to enhance human–machine systems. As this field evolves, there is an increasing emphasis on protective functions, visibility, accessibility, and comfort in engineering machinery operations. The core drilling rig, a sophisticated geotechnical drilling apparatus, comprises various motion mechanisms, adding complexity to its operational system [4]. The driller’s cabin, an advanced control center, manages the core drilling rig and facilitates safety information. Ergonomic validation in this context aims to refine the internal structure and layout of the cabin, thereby reducing labor intensity and improving drilling efficiency and quality.

In the product development stage, traditional ergonomic evaluation typically adopts physical models. Bazazan et al. [5] utilized questionnaires and rapid upper limb assessments to compare operators’ perceptions and workloads in varying postures. However, this method is often time-consuming, expensive, and inflexible. The integration of virtual human technology in ergonomic design overcomes the reliance on physical equipment, offering a realistic multi-sensory virtual experience for effective performance evaluations [6]. This approach addresses the challenges in data collection, working condition simulation, and safety risks. Recent studies [7,8,9] have developed virtual human models and simulations that closely replicate human characteristics, facilitating 3D behavior simulations in virtual environments for ergonomic assessments in various postures. Nevertheless, these simulations are predominantly static, requiring manual posture adjustments and limiting their realism.

Dynamic interactive virtual human–machine simulations enable operators to adaptively adjust themselves in immersive virtual environments, thereby enhancing the natural interactivity between human and the virtual realm and augmenting the simulations’ reliability and efficiency. This evaluation approach encompasses two types: computer vision and sensor-based equipment. Nadeau [10] utilized cameras to record the working processes of aviation de-icing workers, analyzing the impact of varying postures and different weights of de-icing nozzles on human joints and intervertebral discs. Ma [11] employed electromyography to investigate physical strain, uncovering the potential sources of high workloads. Rolf [12] integrated motion capture technology with surface electromyography to evaluate driving posture comfort. Caputo [13] applied motion capture to analyze limb movements, thereby enhancing work comfortability and efficiency through the assessment of limb movement angles. Gong [14] applied the real-time motion data from wearers to the joints of robots, facilitating direct and natural remote control of the robot’s movements by the wearer. These techniques enable the virtual human model to be driven in real-time by the operator via a comprehensive motion capture system, transforming the virtual entity into a motion avatar of the operator, thus acquiring authentic work data from workers, and rendering the evaluation outcomes more accurate and trustworthy.

This study addresses the design of the driller’s cabin in a core drilling rig, introducing a computer-assisted method to evaluate the upper limb comfortability of drillers using digital human technology. It leverages inertial motion capture sensors to simulate the virtual driller’s actions based on real human movements, resulting in more precise and realistic human–machine analysis data. The study establishes a simplified human upper limb model for calculating forces and torques on the joints, thereby assessing the rationality of the virtual driller’s cabin design. This approach offers more authentic and reliable evaluation results for the driller’s cabin design, significantly contributing to the design and enhancement of engineering machinery equipment.

## 2. Digital Human Model Generation

### 2.1. Establishment of Three-Dimensional Virtual Human Model

Three-dimensional virtual human models are digitized representations in human–machine engineering simulations, which are adept at accurately imitating various human postures and facilitating the simulation of human behaviors in three-dimensional environments. Primarily, these models assess product usability, ergonomic design, and comfortability [15]. The human–machine efficiency software, Jack version 8.0.1, offers precise biomechanical human models, enabling the customization of digital human dimensions for the creation of highly realistic virtual humans. Utilizing inverse kinematics for model adjustment, Jack version 8.0.1 simplifies the process, requiring modifications only at crucial points instead of every joint. The model is developed with the 95th percentile adult Chinese male as the reference standard, and its key parameters are detailed in Table 1.

For subsequent bone rigging in Blender version 4.0 software, the model is subjected to format conversion, refined by removing superfluous vertices and faces, and completed with detailed facial data, which culminates in an optimized human body model, as depicted in Figure 1.

### 2.2. Motion Capture Device-Driven Generation of a Digital Human

The motion of the digital human body is orchestrated by a sequence of skeletal movements, subsequently influencing the dynamics of the skin network. To ensure the computational accuracy of motion capture data, aligning the motion capture’s driving points with the virtual human’s joint points is essential, which necessitates constructing a skeletal model. In this study, the VDsuit-Full full-body motion capture system (VIRDYN, Guangzhou, China) is employed (Figure 2), equipped with 27 highly sensitive sensor nodes capable of capturing and outputting data across a wide range of postures. In terms of the accuracy of the device applied to ergonomic assessment, the attitude accuracy of the VDsuit-Full system is the ROLL < 0.5°PITCH < 0.5°YAW < 1°; the acceleration parameter is ±8 G; the gyroscope parameter is ±2000; and the magnetometer parameter is ±4.9 Gs. This is similar to the Perception Neuron Inertial Mo-Cap system of the (NOITOM, Beijing, China). By comparing the accuracy evaluation with the gold-standard (ISO 13485:2016) [16] Vicon system (Vicon Motion Systems Ltd., Oxford, UK), its error is within 5° [17]. The assessments were based on ergonomic criteria, particularly ISO 11228-1:2003 (standard for manual operation) and ISO 9241-210:2019 (user-centric design standard for human-computer interaction), following the correct posture and movement guide [18,19]. This shows that the system is suitable for the accuracy of ergonomic evaluation.

Based on the structural characteristics of the human body, its various components can be organized into a tree-like system. In this system, the pelvic body segment serves as the root node, while the head, hands, and feet function serve as endpoints [20]. Utilizing Blender version 4.0’s skeletal system, a skeleton is constructed by linking the model’s vertices to the skeletal structure, enabling the model to undergo deformation in response to skeletal movements. The core of the skeleton comprises 23 segments, and the hierarchical and skeletal structures of this core are illustrated in Figure 3, with the corresponding hierarchical structure of the upper limb skeleton depicted in Figure 4. 

Digital human movement includes both skeletal motion and skin mesh model deformation. To this end, this study associates the skin mesh vertices with adjacent skeletal segments, establishing a weight relationship between the skin and the skeleton. The skeleton undergoes automatic weight binding, followed by a manual check and adjustment in posture mode to align the skeleton with the mesh model accurately [21]. To ensure the wearer’s movements align with those of the virtual human, motion capture data typically undergo reorientation. This reorientation is achieved by replicating human movements, as collected by the sensors, on the skeletal model. See Figure 5.

## 3. Evaluation of Human Comfortability

### 3.1. Calculation of Joint Angle in the Human Upper Limb 

To address the challenge of setting a consistent zero point in testing, this paper utilizes the spatial vector method for defining points’ spatial positions, aiming to reduce errors arising from zero-point discrepancies. It adopts the left hand as a standard and simplifies the human upper limb into a ball-and-stick model, with the definitions for upper limb joint angles illustrated in Figure 6.

In this model, Points A, B, C, and D correspond to the spatial positions of the upper arm, lower arm, hand, and finger skeletal nodes, respectively. The shoulder joint angle α is defined as the angle between the vertical line at Point A and the line connecting Points A and B. The elbow joint angle β is the angle formed by the extension line of the line from Point A to B and the line from Point B to C. The wrist joint angle γ is determined by the angle between the extension line of the line from Point B to C and the line from Point C to D. A, B, C, and D are all spatial points. θ_1_, θ_2_, and θ_3_ are the angles between the shoulder, elbow, and wrist joints and the horizontal direction, respectively. Taking the angle between the upper arm, lower arm, and hand as an example, the angle between the upper arm and lower arm is δ, the angle between the lower arm and hand is φ. The angle can be calculated using the cosine theorem, and the expression is as follows: (1)$$\left\{\begin{array}{c} \cos\delta =\frac{AB\times BC}{\left|AB|\times |BC\right|}\\ \cos\varphi =\frac{BC\times CD}{\left|BC|\times |CD\right|} \end{array}\right.$$
(2)$$\left\{\begin{array}{c} \theta_{1}=\pi /2-\alpha \\ \theta_{2}=\pi -\theta_{1}-\delta =\beta -\theta_{1}\\ \theta_{3}=\pi +\theta_{2}-\varphi =\gamma +\theta_{2} \end{array}\right.$$

Based on the disparity in spatial node coordinates within the bone structure, the respective position vectors are derived. Subsequently, Equation (1) is applied to determine the angles formed by the shoulder, elbow, and wrist joints during upper limb movement. Building upon Equation (1), the values of θ_1_, θ_2_, and θ_3_ are acquired through the angle conversion specified in Equation (2). Utilizing the obtained joint angle data, MATLAB version R2020a is employed to compute the mean, standard deviation, maximum, and minimum values for each joint angle.

### 3.2. Calculation of Upper Extremity Joint Moments

Evaluating human comfortability exclusively through joint angles is inadequate for precisely uncovering the actual sources of discomfort. The calculation of the joint torques provides an intuitive assessment of joint loading, offering a more precise evaluation of joint comfortability [22]. In the analysis of the upper limb moment, considering the hand’s relatively minor mass, the lower arm is the main body for the mass position. Consequently, wrist joint force analysis is omitted, with attention directed towards the potential load impact on the lower arm. The stress analysis of the human upper limbs during exercise is shown in Figure 7.

In this analysis, point A signifies the shoulder joint’s center, B represents the upper arm’s center of mass, C is the elbow joint’s center, and D indicates the center of mass of the forearm and hand. G_1_ and G_2_ are the gravitational forces acting on the forearm and hand, and the upper arm, respectively. R_S_ is the reaction force at the shoulder joint’s center generated by the entire upper limb. R_E_ and R_E’_ represent interactive forces at the elbow joint during upper limb movement. R_p_ and R_p’_ represent interactive forces at the wrist joint during hand movement. θ_E_ and θ_S_ are the angles of the lower arm and upper arm relative to the horizontal plane, respectively. F_t1_ and F_r1_ represent the tangential and radial forces, respectively, when the forearm and hand move around the elbow joint. F_t2_ and F_r2_ represent the tangential and radial forces, respectively, when the upper arm moves around the shoulder joint. F_t0_ and F_r0_ are the tangential and radial forces, respectively, when the forearm and hand move around the shoulder joint. L_0_, L_1_, and L_2_ represent the distances from the shoulder joint center to the forearm center of mass, from the elbow joint center to the forearm center of mass, and from the center of mass of the upper arm to the shoulder joint center, respectively. M_S_ is the torque of force corresponding to the entire upper limb at the shoulder joint A, M_E_ is the torque of force corresponding to the forearm at the elbow joint C, and M_p_ is the torque of force corresponding to the hand at the wrist joint E. For the convenience of subsequent calculations, it is assumed that the mass per unit length of the arm is m_0_, with specific calculations as follows:(3)$$\overset{..}{\theta }=\frac{d\omega }{dt}=\frac{d\left(\frac{d\theta }{dt}\right)}{dt}=\frac{d^{2}\theta }{dt^{2}}$$
(4)$$mR=\int_{0}^{L}m_{0}rdr$$
(5)$$F_{t}=mR\overset{..}{\theta }$$
(6)$$F_{r}=m{\left(\frac{d\theta }{dt}\right)}^{2}R=m\frac{d\theta^{2}}{dt^{2}}R$$

Based on this, the formulation of inverse dynamic equilibrium equations and torque balance equations for the elbow joint and shoulder joint is presented. According to the principles of kinematics, it is known that the radial forces generated by the upper limb movements are all on the extension line of the limb, so radial forces do not produce moments at the joints. Equation (7) below represents the force and torque balance equations for the forearm, and Equation (8) represents the force and torque balance equations for the upper arm.
(7)$$\left\{\begin{array}{c} \vec{M_{E}}+\vec{G_{1}}\times L_{1}\times \cos\theta_{E}+\vec{F_{t1}}\times L_{1}=0\\ \vec{R_{E}}+\vec{{R_{p}}_{’}}+\vec{G_{1}}+\vec{F_{t1}}\times \cos\theta_{E}+\vec{F_{r1}}\times \sin\theta_{E}=0 \end{array}\right.$$
(8)$$\left\{\begin{array}{c} \vec{M_{S}}+\vec{G_{2}}\times L_{2}\times \cos\theta_{S}+\vec{F_{t2}}\times L_{2}+\vec{F_{t0}}\times L_{0}=0\\ \vec{R_{S}}+\vec{R_{E’}}+\vec{G_{2}}+\vec{F_{t2}}\times {\cos\theta }_{S}+\vec{F_{r2}}\times {\sin\theta }_{S}=0 \end{array}\right.$$

Data processing and analysis were conducted using MATLAB version R2020a software, enabling the calculation of torque variation curves for both the elbow and shoulder joints. These curves were graphically represented with frame number as the variable.

## 4. Experiment 

### 4.1. Data Acquisition

In the initial phase of the study, Jack version 8.0.1 and CATIA V5 software version V5-6R2022 were employed to design the driller’s cabin for the Chinese 5000 m core drilling rig and to verify its ergonomics. Figure 8 depicts the finalized design of the driller’s cabin. To evaluate the design’s efficacy, motion capture sensor data were analyzed. A virtual simulation platform for assessing man-machine efficiency was developed using Unity3D. When integrating real-time motion capture data in Unity3D, it is necessary to pay attention to hardware performance, data processing load and software scalability, optimize the process such as adopting multi-threading technology, and consider additional plugins or custom development to meet complex requirements to ensure data accuracy and real-time performance. In the process of construction, resources were added to the scene by adding the plane model and lights provided by Unity3D, and position, rotation, and scale data were adjusted in the transform window. If the spatial layout of the entire scene is not considered in the above operations, it may lead to overlap between resources, affecting the visual effect and interaction design. Similarly, there is also a parent–child relationship between resources, in order to prevent layout confusion, it is necessary to frequently test and preview the effect during the adjustment process to ensure that it meets the design requirements. After importing the driller’s room model and the character model with skeleton into Unity3D, real-time action data were broadcast to Unity3D using the VDMocap Studio software version 2.1.15 provided by the hardware vendor. Note that when motion capture data are transmitted in real time and synchronized with the 3D model, this integration may face a processing overload, especially if high frequency sampling and complex calculations are required. In order to avoid delay or frame loss, it can be solved by reducing unnecessary calculations and improving hardware performance.

To improve the realism of the driller’s cabin and digital human within the virtual platform, collision detection functionality was integrated between the cabin and human models. This integration involved the creation of bounding box components for rigid bodies, coupled with the use of inherent triggers to identify overlaps between collision boxes. The Rokoko Studio for Blender plugin (Rokoko, San Francisco, CA, USA) facilitated the alignment and redirection of skeletons using data exported from VDMocap Studio (VIRDYN, Guangzhou, China). Ultimately, the process culminated in the export of files encompassing the human mesh model, the skeletal system with bindings, associated materials and textures, and skeletal motion data. The overall simulation experimental testing platform is shown in Figure 9.

The driller predominantly engages in three actions: manipulating the joystick, adjusting the knob, and activating buttons. Data on these key actions are systematically collected. Throughout the testing phase, the digital human’s movements are visualized via animation frame playback. Within Unity3D, the transform component systematically captures the skeletal nodes’ numerical values on a frame-by-frame basis to gather positional data. Both the data collection and subsequent processing are conducted at a consistent frame rate of 60 frames per second. We selected five adult males who roughly met 95% of the criteria in this paper for operational testing. Before the data collection, informed consent was given by the participants. During the testing process, we tried our best to keep the testers at the same zero point of the initial state. After normal operation testing, due to the basically consistent sequence of operation process, and after data collection, it was found that the position data were roughly the same and did not vary too much in the process of interception in seconds. In addition, in order to make the research of test data universally representative, we intercepted the comprehensive average data of the upper limb movements of the five testers. Table 2 displays the average position data of skeletal nodes obtained during a random period of time.

### 4.2. Calculation of Upper Extremity Joint Angles and Torques

Upon collecting positional data from three typical arm movements, five key parameters are computed: the angles of the shoulder, elbow, and wrist joints, along with the torques at the shoulder and elbow joints. The variation curves for these parameters are depicted in Figure 10, Figure 11 and Figure 12. Table 3 displays the average, maximum, and minimum value of the upper limb joint angles and torques for these movements.

### 4.3. Assessment Criteria

To demonstrate the accuracy of the model design, the operational comfort of the drillers in this study is evaluated next. There are many evaluation modes, and one of the average modes is selected to evaluate the comfort of this design [23]. According to the ergonomic design of geological core rig, the joint angles of the human body have limited angle range and comfortable angle range under different activity modes [24,25]. The angle adjustment range for the comfortable posture of different body parts is shown in Table 4.

For force comfort, the human upper limb comfort evaluation is itself a subjective mapping of the upper limb muscle strength, which can be assessed by combining the subjective feelings of the tester and the classification of the upper limb muscle comfort grade provided in the literature [26]. The corresponding muscle strength X_i_ is obtained by converting the torque. Because the increase in X_i_, muscle fatigue is inversely proportional to comfort, the muscle strength X_i_ is input into the following upper limb muscle comfort function:$$\sigma =\frac{\int_{0}^{t}(X_{\max}-X_{i})dt}{\int_{0}^{t}(X_{\max}-X_{i^{’}})dt}\;\left(i\in N^{*}\right)$$

In this analysis, X_max_ is the maximum muscle strength value calculated above, X_i_ represents the muscle strength of the joint under test by the tester during the simulation operation, and X_i’_ represents the muscle strength of the joint under test in the natural state under the same continuous frame number as the simulation operation. In accordance with the above process, the average comfort index σ of the experimental test data is shown in the Table 5.

Then, the force comfort is evaluated effectively, as shown in Table 6 and demonstrated in the literature [26].

### 4.4. Analysis of Visible and Reachable Domains

The analysis of the visual domains is grounded in the eye ellipsoid principle. This study employs the Unity3D camera system to emulate the human eye’s visual cone, ensuring a consistent relative positioning of both the visual cone and the visual field concerning the head skeleton. By aligning the camera’s visual cone and the visual field with the virtual eye’s position, this study defines the scope of vision, as demonstrated in Figure 13.

The tracking of the human hand bone position is conducted, with its positional data recorded in each frame, and line segments are rendered using the Unity3D engine. Analyzing the spatial relationship between the hand trajectory curve and the manipulation device model’s range enables reachable range analysis, as depicted in Figure 14.

## 5. Discussion

By combining motion capture technology with real-time data analysis, the study achieves accurate capture and just-in-time analysis of joint angle and torque in a highly dynamic environment, which provides strong support for ergonomic improvements in complex work scenarios and reveals more design details. The experimental findings indicate that the elbow joint angle is typically largest, followed by the shoulder joint angle, and the wrist joint angle is smallest. A smooth angle change curve indicates there is no movement variation during this period, with the joint angle remaining constant and the upper limb acceleration at zero, bearing only gravitational torque. This phenomenon is corroborated by the torque curve diagram.

The data (Table 3 and Table 4) reveal that, during three typical movements, the average values for both the shoulder and elbow joints fall within the comfort range, and the wrist joint’s average value is closely aligned with this zone. Notably, the maximum and minimum value for the angles of the shoulder, elbow, and wrist joints all fall within the human body’s range of motion limits. Similarly, the data (Table 5 and Table 6) reveal that the force comfort index of shoulder and elbow is also above the “more comfort” level, which indicates that the operation comfort of the experimental test meets the design requirements of the core rig.

This experiment employs a virtual driller’s cabin model designed using Jack version 8.0.1 software and performs an ergonomic evaluation with Catia software version V5-6R2022. The study utilizes motion capture sensors to create a digital human, enabling the quantitative analysis of upper limb comfort, visual field, and reachable range, thereby affirming the validity of the previous cabin design.

## 6. Conclusions

Traditional ergonomic evaluation methods are challenged by lengthy processes, high cost, and lack of flexibility in adjustment. Virtual ergonomics and digital human modeling are pivotal in digital manufacturing and product development and advancing swiftly. This paper introduces a digital human-based method for ergonomic analysis in the driller’s cabin of core drilling. Operations are conducted through inertial motion capture sensors, the virtual drillers are controlled by the real human operators, and the operations yield more precise and lifelike human–machine analysis data. Concurrently, the study establishes a model of human upper limb to calculate the force and torque of upper limb joints and quantitatively evaluate the rationality of driller room design. This study also provides a more nuanced analysis of joint angle and torque than the simplified treatment of human posture in many existing studies. This depth analysis not only improves the accuracy of the data but also quantitatively evaluates the rationality of the driller room design. The study makes a significant contribution to the integration of digital human technology with dynamic and interactive virtual human–machine simulation, thereby enhancing the operator’s natural interaction with the virtual environment and augmenting the simulation’s reliability and efficiency. Digital human technology also addresses challenges like the difficulties in data collection and the complexity of simulating working conditions. In future research, the application of digital human technology can be further explored in ergonomics evaluation. For example, higher-precision sensors can be integrated or more complex algorithms developed to achieve more accurate, realistic digital human simulations. Similarly, in the face of highly dynamic and complex environments in aviation, automotive, and other fields, based on the case study in this paper, stable operation performance under extreme conditions can be improved through the combination of motion capture technology and real-time data analysis, and the adaptability of the system can be improved by providing real-time analysis results. Ergonomics involves the cross-fusion of many fields, and the application in these fields requires deep integration and collaborative work with these technologies to achieve the best ergonomic evaluation effect. While the direct impact of this approach has been analyzed in this paper, its long-term impact on ergonomic design and human health remains unexplored. Future research should investigate the sustainability of this approach applied over time, assessing how continued use may affect ergonomic outcomes and contribute to overall health and well-being, and should improve the adaptability of the system in different environments to achieve more accurate and efficient ergonomic evaluations.

## Figures and Tables

**Figure 1 sensors-24-05962-f001:**
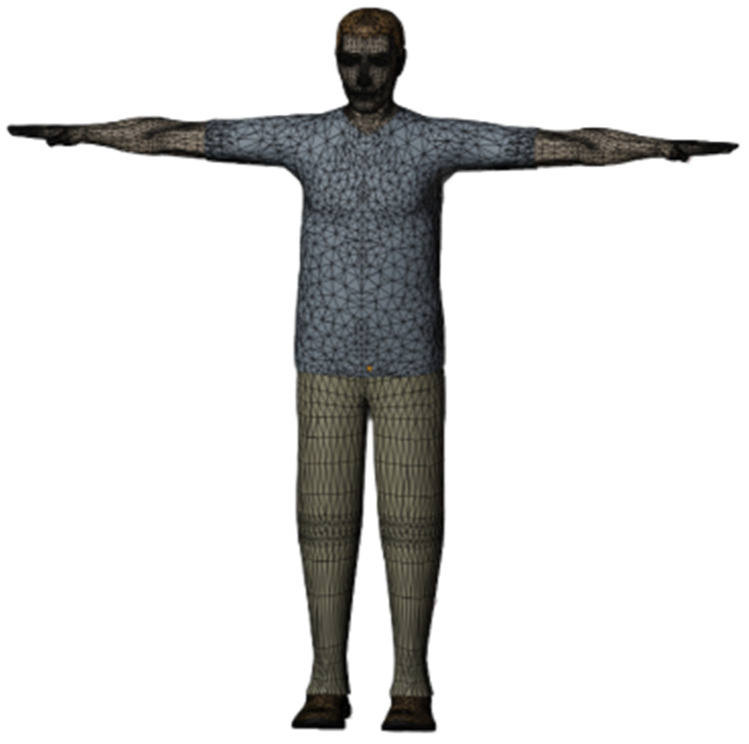
Virtual human model.

**Figure 2 sensors-24-05962-f002:**
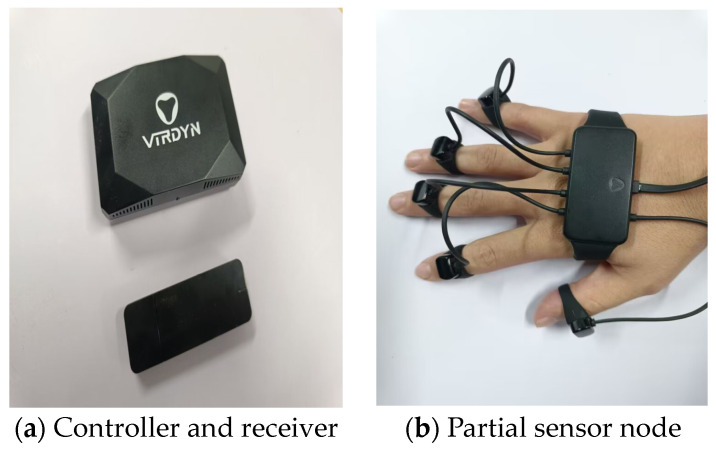
Motion capture sensor and wear effect.

**Figure 3 sensors-24-05962-f003:**
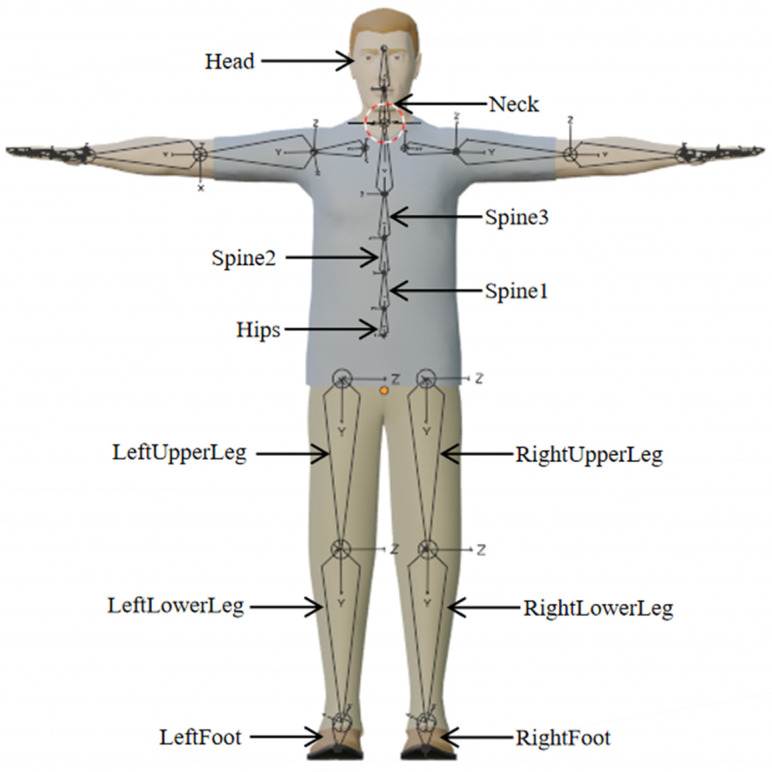
Parent–child relationship of human body skeleton.

**Figure 4 sensors-24-05962-f004:**
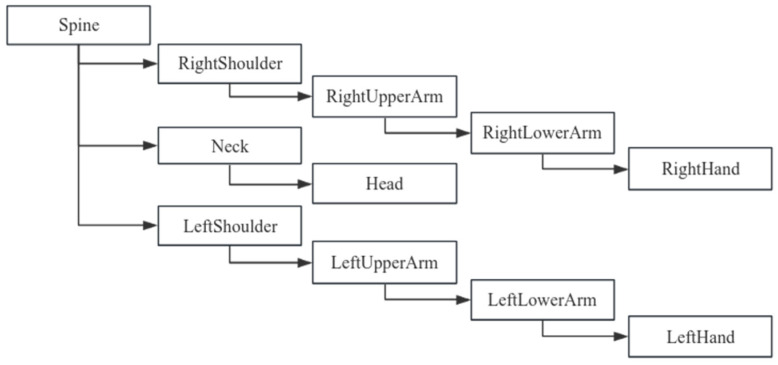
Parent–child relationship of human upper limb bones.

**Figure 5 sensors-24-05962-f005:**
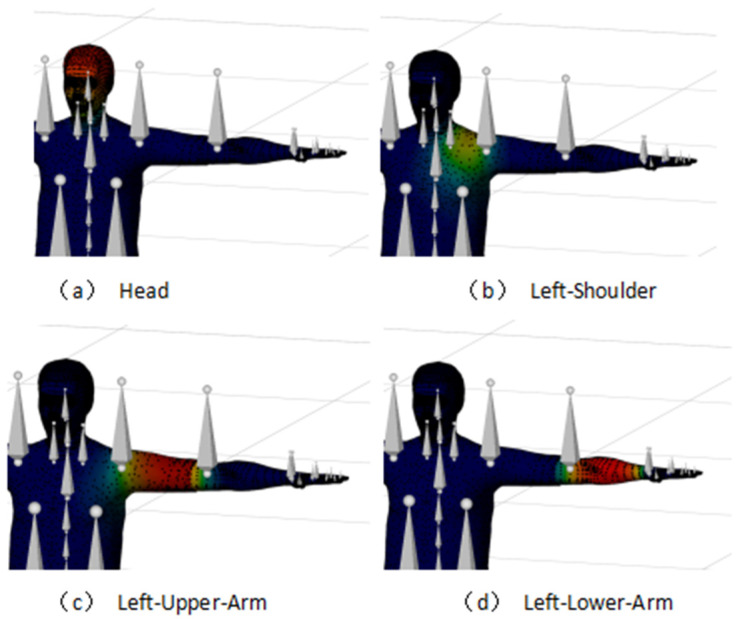
Weight of the partial bones in the skeleton.

**Figure 6 sensors-24-05962-f006:**
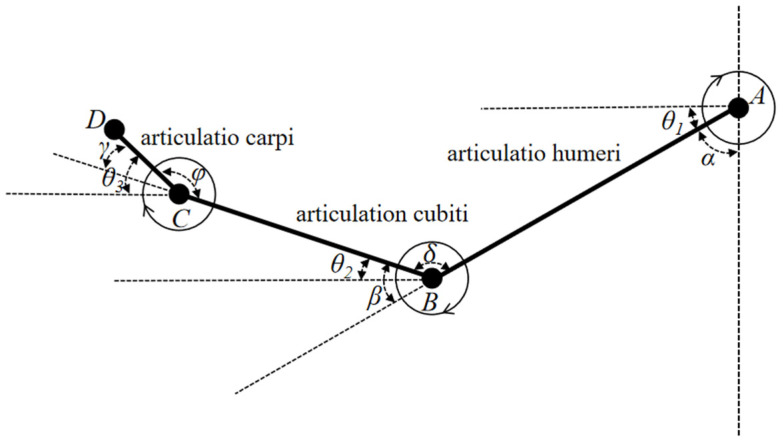
Angle analysis diagram of upper limb joint.

**Figure 7 sensors-24-05962-f007:**
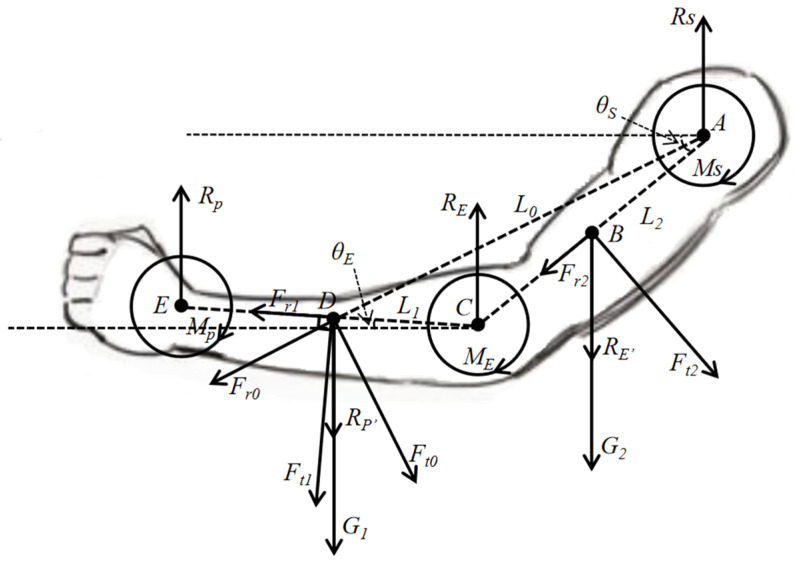
Torque analysis diagram of upper limb joints.

**Figure 8 sensors-24-05962-f008:**
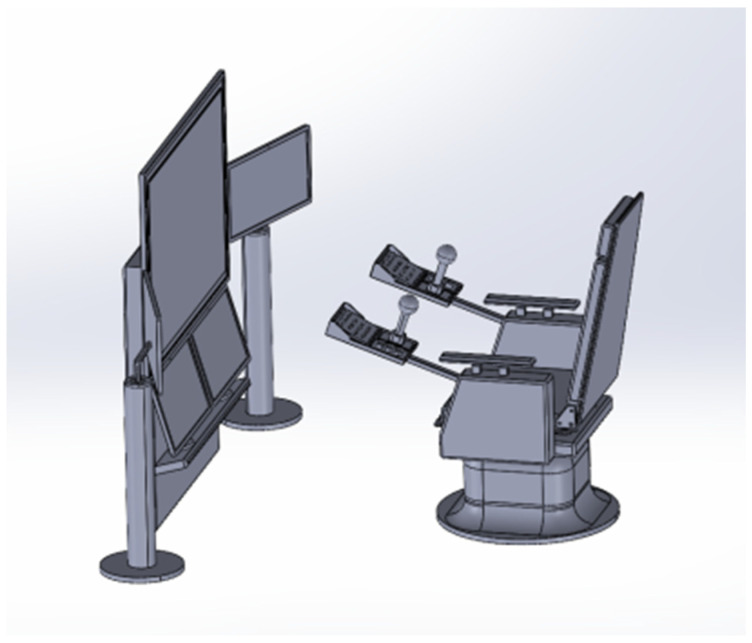
The 3D structure of the core drilling driller’s cabin.

**Figure 9 sensors-24-05962-f009:**
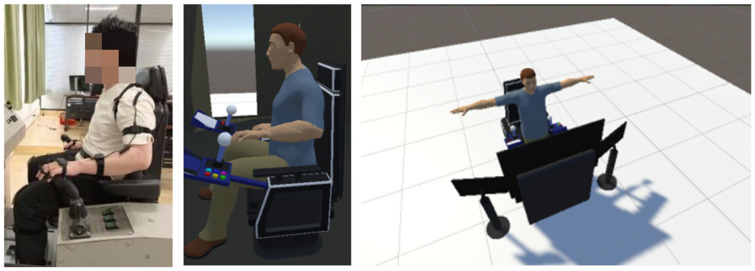
Test platform.

**Figure 10 sensors-24-05962-f010:**
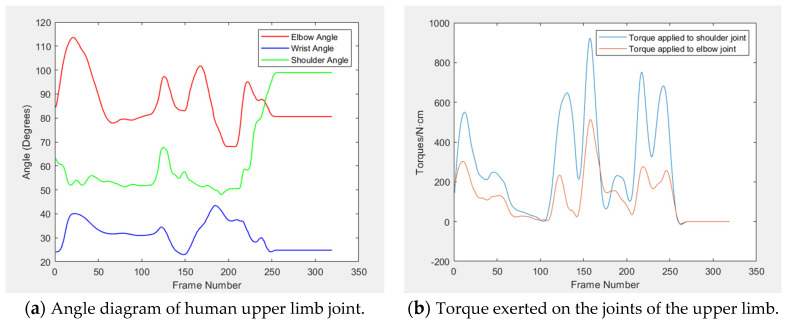
Curves of joint angle and torque under operating rocker.

**Figure 11 sensors-24-05962-f011:**
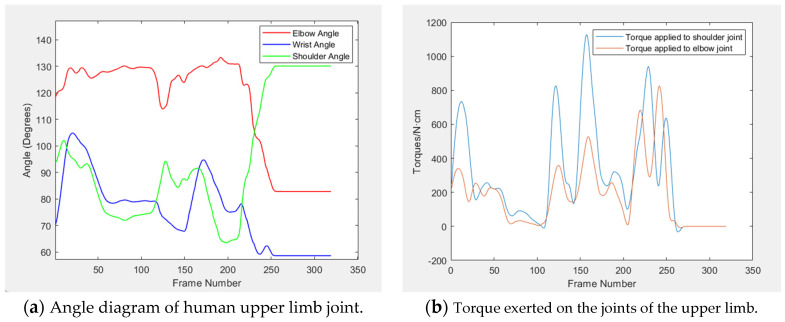
Curves of joint angle and torque under twist knob.

**Figure 12 sensors-24-05962-f012:**
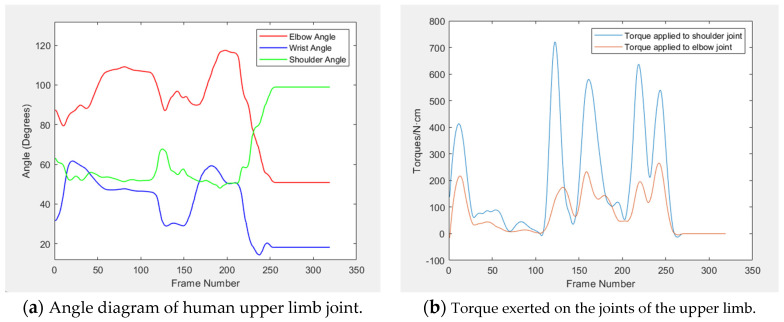
Curves of joint angle and torque under push button.

**Figure 13 sensors-24-05962-f013:**
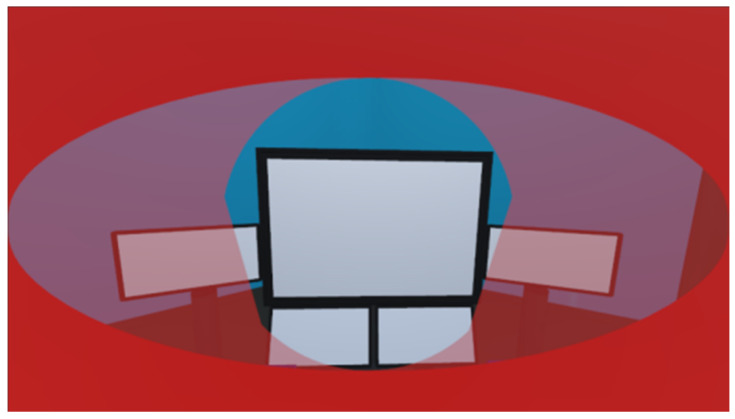
Analysis of visible domains.

**Figure 14 sensors-24-05962-f014:**
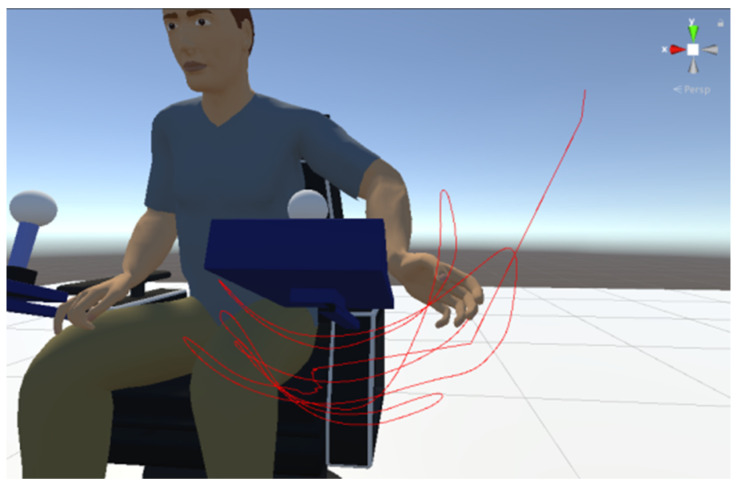
Analysis of reachable domains.

**Table 1 sensors-24-05962-t001:** Main parameters of human body.

Anthropometry Parameters/Human of the 95th Percentile (cm)								
Stature	Weight	Head Length	Acromion Height	Biacromial Breadth	Arm Length	Elbow Span	Buttock-Popliteal Length	Thigh Clearance
177.5	75.0	19.8	147.1	39.7	79.9	135.0	49.0	16.2

**Table 2 sensors-24-05962-t002:** Random position data at 60 frames per second.

Position Data of Each Node at the Same Time			Random Sampling Node Number									
			1	2	3	4	5	6	7	8	9	10
Operating rocker	Left Hand	x	−0.124	−0.122	−0.118	−0.112	−0.108	−0.103	−0.098	−0.093	−0.090	−0.089
		y	0.843	0.847	0.854	0.864	0.873	0.884	0.896	0.910	0.925	0.938
		z	0.194	0.192	0.191	0.189	0.188	0.186	0.187	0.189	0.193	0.199
	Left Lower Arm	x	−0.126	−0.123	−0.119	−0.114	−0.110	−0.105	−0.100	−0.095	−0.092	−0.091
		y	0.839	0.843	0.849	0.859	0.867	0.878	0.890	0.904	0.919	0.931
		z	0.188	0.186	0.184	0.183	0.182	0.181	0.181	0.184	0.189	0.194
	Left Upper Arm	x	−0.292	−0.292	−0.292	−0.293	−0.294	−0.295	−0.297	−0.299	−0.301	−0.303
		y	0.931	0.930	0.930	0.929	0.930	0.930	0.931	0.933	0.935	0.938
		z	0.011	0.010	0.009	0.009	0.010	0.013	0.018	0.026	0.035	0.044
	Left Shoulder	x	−0.183	−0.184	−0.184	−0.184	−0.184	−0.184	−0.184	−0.184	−0.184	−0.184
		y	1.175	1.174	1.173	1.173	1.173	1.172	1.172	1.172	1.172	1.172
		z	−0.001	−0.002	−0.002	−0.003	−0.004	−0.004	−0.005	−0.005	−0.005	−0.005
Twist knob	Left Hand	x	−0.092	−0.092	−0.091	−0.091	−0.091	−0.092	−0.092	−0.093	−0.093	−0.093
		y	0.836	0.836	0.837	0.837	0.838	0.836	0.839	0.839	0.839	0.840
		z	0.034	0.034	0.038	0.039	0.041	0.042	0.043	0.044	0.045	0.045
	Left Lower Arm	x	−0.092	−0.092	−0.092	−0.092	−0.092	−0.092	−0.092	−0.092	−0.092	−0.093
		y	0.831	0.831	0.831	0.832	0.832	0.833	0.833	0.833	0.834	0.834
		z	0.028	0.028	0.032	0.033	0.035	0.036	0.037	0.038	0.039	0.040
	Left Upper Arm	x	−0.231	−0.231	−0.230	−0.230	−0.229	−0.229	−0.228	−0.228	−0.228	−0.229
		y	0.929	0.929	0.928	0.927	0.927	0.927	0.927	0.927	0.926	0.926
		z	−0.166	−0.166	−0.164	−0.163	−0.163	−0.162	−0.162	−0.161	−0.161	−0.160
	Left Shoulder	x	−0.136	−0.136	−0.136	−0.135	−0.135	−0.135	−0.135	−0.134	−0.134	−0.134
		y	1.166	1.166	1.166	1.166	1.166	1.165	1.165	1.165	1.165	1.165
		z	−0.090	−0.090	−0.091	−0.091	−0.090	−0.090	−0.090	−0.090	−0.090	−0.090
Push button	Left Hand	x	−0.359	−0.360	−0.360	−0.359	−0.359	−0.358	−0.356	−0.354	−0.352	−0.349
		y	0.868	0.868	0.869	0.869	0.869	0.869	0.869	0.870	0.870	0.871
		z	0.198	0.201	0.202	0.204	0.204	0.204	0.205	0.205	0.205	0.206
	Left Lower Arm	x	−0.357	−0.357	−0.358	−0.357	−0.356	−0.355	−0.353	−0.351	−0.349	−0.347
		y	0.863	0.863	0.864	0.864	0.864	0.865	0.865	0.866	0.866	0.866
		z	0.192	0.194	0.196	0.197	0.198	0.198	0.198	0.198	0.198	0.199
	Left Upper Arm	x	−0.296	−0.297	−0.297	−0.298	−0.298	−0.298	−0.299	−0.299	−0.299	−0.301
		y	0.974	0.975	0.976	0.976	0.977	0.977	0.976	0.975	0.975	0.975
		z	0.011	0.010	0.009	0.009	0.010	0.013	0.018	0.026	0.035	0.044
	Left Shoulder	x	−0.128	−0.128	−0.128	−0.129	−0.129	−0.129	−0.130	−0.130	−0.130	−0.131
		y	1.171	1.171	1.172	1.173	1.173	1.172	1.172	1.172	1.172	1.172
		z	−0.097	−0.097	−0.097	−0.096	−0.096	−0.095	−0.095	−0.096	−0.094	−0.094

**Table 3 sensors-24-05962-t003:** Data processing results during the three actions.

Upper Limb Movement	Joint Parameter	Experimental Data Statistics		
		Average Value	Minimum Value	Maximum Value
Operating rocker	Shoulder angle (°)	66.04	48.01	98.91
	Elbow angle (°)	86.32	67.98	113.62
	Wrist angle (°)	31.02	22.99	43.47
	Shoulder torques (N·cm)	253.39	0	857.79
	Elbow torques (N·cm)	119.66	0	483.62
Twist knob	Shoulder angle (°)	95.46	65.79	130.06
	Elbow angle (°)	108.65	82.81	133.29
	Wrist angle (°)	78.82	58.12	104.85
	Shoulder torques (N·cm)	296.78	0	1108.45
	Elbow torques (N·cm)	189.35	0	746.83
Push button	Shoulder angle (°)	66.04	48.01	98.91
	Elbow angle (°)	82.61	50.83	117.43
	Wrist angle (°)	36.30	14.32	63.18
	Shoulder torques (N·cm)	213.80	0	738.64
	Elbow torques (N·cm)	101.33	0	250.21

**Table 4 sensors-24-05962-t004:** Postural and angle range of shoulder, elbow, and wrist joints.

Joint	Mode of Motion	Limiting Angle	Comfort Zone
Shoulder joint	Front and rear pendulum	140°~40°	40°~90°
Elbow joint	Bend and stretch	140°~40°	80°~110°
Wrist joint	Wrist flexion and extension	80°~70°	10°~30°

**Table 5 sensors-24-05962-t005:** Average comfort index under the three movements.

Joint	Upper Limb Movement		
	Operating Rocker	Twist Knob	Push Button
Shoulder joint	0.782	0.758	0.800
Elbow joint	0.833	0.797	0.815

**Table 6 sensors-24-05962-t006:** Comfort rating table.

Comfort Level	I	II	III	IV	V
Comfort index	0≤σ<0.2	0.2≤σ<0.4	0.4≤σ<0.6	0.6≤σ<0.8	σ≥0.8
Comfort description	Very uncomfortable	Not comfortable	Generally comfortable	More comfortable	Very comfortable

## Data Availability

The original contributions presented in the study are included in the article, further inquiries can be directed to the corresponding author.

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
