# Peer review of "Inertial Motion Capture-Driven Digital Human for Ergonomic Validation: A Case Study of Core Drilling"

_sensors, 2024, doi:10.3390/s24185962_

Round 1

Reviewer 1 Report

Comments and Suggestions for Authors

In this study, the authors present research utilizing a computer-assisted method for evaluating the upper limb comfort of drillers. They employed the VDsuit-Full full-body motion capture system to record the subjects' movements. After reviewing this manuscript, several questions have arisen. I request the authors to provide further information or comments on the following points:

1)The VDsuit-Full full-body motion capture system can be used to generate three-dimensional computer animations in real-time. I do not question the capabilities of this system. However, since the authors intend to apply it in ergonomic studies, it is essential to confirm the system's accuracy, possibly by comparing it with a gold standard. If such data has already been reported in the literature, the authors should address and discuss this in their manuscript.

2)This study only reports on a single case. In biomedical research, only very rare or unique cases warrant publication as a case report. I suggest that the authors recruit additional subjects to strengthen the experimental results.

3)Regarding the ergonomic evaluation, how do the authors assess compliance with standards related to joint angles and torques?

Author Response

Response to Reviewer X Comments

1. Summary

We appreciate the insightful comments provided by the reviewer, which have significantly contributed to the enhancement of our manuscript. The reviewer raised several critical points, including the need for a more thorough validation of the VDsuit-Full motion capture system against gold standards, the importance of expanding our subject pool, and the necessity of providing more detailed explanations regarding the challenges and limitations of using Unity3D as a software tool. Additionally, the reviewer emphasized the need to more clearly link our study's findings to broader ergonomic implications and to better ground future research directions in the current study's results and limitations.

In response, we have made several revisions to address these concerns. We illustrate the vdsuit-full system validation process by citing the literature and discuss its accuracy in the optical system with the gold standard Vicon. We clarified our approach to data selection and provided justification for the representative analysis used. Furthermore, we expanded the discussion on the ergonomic evaluation criteria and included additional context about the challenges encountered when integrating real-time motion data with Unity3D. Lastly, we refined the conclusion to more explicitly connect our findings to future research directions, underscoring how our results guide further studies and addressing the limitations observed in our current work.We believe these revisions have strengthened the manuscript, providing a more comprehensive and detailed account of our research methodology, results, and implications.

2. Point-by-point response to Comments and Suggestions for Authors

Comments 1: The VDsuit-Full full-body motion capture system can be used to generate three-dimensional computer animations in real-time. I do not question the capabilities of this system. However, since the authors intend to apply it in ergonomic studies, it is essential to confirm the system's accuracy, possibly by comparing it with a gold standard. If such data has already been reported in the literature, the authors should address and discuss this in their manuscript.

Response 1: For the VDsuit-Full inertial body motion capture equipment, the accuracy evaluation can be referred to the paper: Validity of the Perception Neuron inertial motion capture system for upper body motion analysis, Measurement,  Volume 149, 2020, 107024, ISSN 0263-2241. In this paper, the Perception Neuron Inertial Mo-Cap system developed by China NOITOM Co., Ltd. is compared with the gold standard Vicon system to evaluate the effectiveness of the Mo-Cap system for estimating the upper body attitude kinematics. As both inertial motion capture equipment, VDsuit-Full inertial body motion capture equipment has attitude accuracy RoLL<0.5°PITCH<0.5°YAW<1°; The acceleration parameter is ±8G. Gyroscope parameters are ±2000; Magnetometer parameters are ±4.9Gs; This is similar to NOITOM's Perception Neuron Inertial Mo-Cap system, which is applied with superior accuracy for ergonomic evaluation. Compared with the gold standard optical whole body motion capture device Vicon system, VDsuit-Full system is an inertia-based whole body motion capture device, and optical motion capture is a motion capture system based on optical tracking marks combined with special cameras, and then data processing and integration. This technology has many advantages, such as high capture accuracy, good real-time performance, easy to wear markers and does not limit the range of human motion, and can collect information for high-speed motion. At present, in the field of optical Motion capture systems, the United States Motion Analysis and the United Kingdom's Vicon (gold standard) are relatively well-known service providers. However, the optical type is easy to be blocked, and is greatly affected by the external environment such as shadows, and it is generally necessary to build a special mobile trap. However, the environment of the drilling room of the core drilling rig in this paper is relatively small and shielded, so this paper mainly uses inertial motion capture equipment to collect the data of the operator's operation movement. The advantages are that the experimental environment is less limited, the acquisition cost is low, the equipment is convenient to wear, and it is not restricted by the occlusion and shadow.(Please see page 4, line 112 for original article modification)

Comments 2: This study only reports on a single case. In biomedical research, only very rare or unique cases warrant publication as a case report. I suggest that the authors recruit additional subjects to strengthen the experimental results.

Response 2: Agree. We did do multiple people to collect data when doing the experiments, To enable the movement data to be generally (certain) representative, The data in the article are based on the composite average data of the upper limb movements of five persons, Because there are 4 points, One is that the tester we selected met roughly 95% criteria for a male sex, And set at the same zero point before starting to reduce the subsequent data processing pressure; And consistent operation action and short acquisition time, After data collection, it is found that there is no excessive floating; Second, in the same number of seconds, The upper limb operation process of different subjects is basically the same; Third, considering the beauty and length influence of the article pictures, And the article focuses on providing a set of methods rather than doing case studies, Therefore, several representative sets of data are selected; Fourth, the method provided in this paper was conducted based on 95% of adult males, Data are representative of the study, And for different personnel, environment test adaptability.(Please see page 10, line 264 for original article modification)

Comments 3: Regarding the ergonomic evaluation, how do the authors assess compliance with standards related to joint angles and torques?

Response 3: Thank you for pointing this out. We agree with this comment. Therefore, We have added a section to the original article, mainly to add the comfort criteria for the assessment angle and torque. The concept of comfort index is introduced. And please see page 13, line 299 for the original article modification.

3. Additional clarifications

Clarification: We would like to provide further details on our decision to use the VDsuit-Full motion capture system and the rationale behind the selection of representative data for analysis. As mentioned earlier, the VDsuit-Full system was chosen due to its practicality and suitability for the confined and obstructed environment of the drilling rig operator's cabin. While not as precise as gold-standard optical systems like Vicon, it offers sufficient accuracy for our ergonomic assessments and is less susceptible to environmental limitations such as occlusion or shadow interference.

Regarding the use of data from five subjects, we chose to present an averaged dataset to represent typical upper limb motion during the tasks. This approach was based on the consistency of the motion data observed across the subjects, which exhibited minimal variation. By focusing on this representative data, we aimed to provide a clear and concise analysis without overwhelming the reader with redundant information.

We hope these additional clarifications address any remaining concerns and provide a deeper certification of our methodological choices and the rationale behind them.

Reviewer 2 Report

Comments and Suggestions for Authors

Novelty and Contribution: The manuscript provides an innovative approach to ergonomic validation but lacks discussion on how it advances beyond existing studies.

The review covers relevant prior work but could benefit from more recent references to contextualize advancements in digital human modeling.

Methodology: The integration of motion capture for real-time data collection is robust, but validation against real-world scenarios is limited. Data Analysis in the analysis of joint angles and torques is thorough; however, the interpretation of results in terms of ergonomic improvement is insufficient.

The paper is well-structured, but some sections, especially the discussion, need clearer linkage between findings and broader ergonomic implications.

The use of software tools like Unity3D is appropriate, yet the manuscript could elaborate more on the challenges and limitations encountered during implementation.

The conclusion suggests future research areas, but these are not sufficiently grounded in the findings or limitations of the current study.

Comments on the Quality of English Language

None

Author Response

Response to Reviewer X Comments

1. Summary

Thank you very much for your thorough and insightful review of our manuscript. We appreciate the time and effort you have taken to provide valuable feedback. Below, we have provided detailed responses to each of your comments, and we have made corresponding revisions to the manuscript. These revisions are highlighted in the re-submitted files.

2. Point-by-point response to Comments and Suggestions for Authors

Comments 1: Novelty and Contribution: The manuscript provides an innovative approach to ergonomic validation but lacks discussion on how it advances beyond existing studies.

Response 1: Thank you for pointing this out. We agree with this comment. Therefore, We have made some revisions to the manuscript. The study combined motion capture technology with real-time data analysis, a relatively rare practice in existing research. Many traditional ergonomic studies may rely on off-line analysis, which is difficult to feedback and adjust in real time. The research is primarily able to collect and process data in real time to provide immediate analytical results, which is particularly important for complex dynamic environments.

Compared with traditional methods in static or low dynamic environments, the method in this study is more adaptable in high dynamic environments. This adaptability is reflected in the ability to accurately capture rapidly changing joint angles and torque, thus providing stronger data support for ergonomic improvements in high-intensity work scenarios, such as the core drill studied in this paper.

Compared with the simplified treatment of human posture in many existing studies, this study provides a more detailed analysis of joint Angle and torque. This in-depth analysis not only improves the accuracy of the data, but also reveals more ergonomic details, which can help design more ergonomic tools and working environments.(Please see page 15, line 350 and page 16, line 373 for the original article modification)

Comments 2: The review covers relevant prior work but could benefit from more recent references to contextualize advancements in digital human modeling.

Response 2: We agree with this comment. We have revised the manuscript to include more recent references that contextualize advancements in digital human modeling. These references provide updated perspectives on the field and enhance the discussion around the latest trends and methodologies in digital human modeling. The revisions can be found on page 17, line 439-457 of the manuscript, where we have incorporated new literature that better situates our study within the current research landscape.

Comments 3: Methodology: The integration of motion capture for real-time data collection is robust, but validation against real-world scenarios is limited. Data Analysis in the analysis of joint angles and torques is thorough; however, the interpretation of results in terms of ergonomic improvement is insufficient.

Response 3: Thank you for pointing this out. We agree with this comment. Therefore, We have added a section to the original article, mainly to add the interpretation of results in terms of ergonomic improvement, and please see page 13, line 299 and page 14, line 309 for the original article modification.

Comments 4: The paper is well-structured, but some sections, especially the discussion, need clearer linkage between findings and broader ergonomic implications.

Response 4: For your valuable advice, in order to more clearly link the results and the wider ergonomics meaning, modified as follows "By combining motion capture technology with real-time data analysis, the study achieves accurate capture and just-of-time analysis of joint angle and torque in a highly dynamic environment, which provides strong support for ergonomic improvements in complex work scenarios and reveals more design details.The experimental findings indicate that the elbow joint angle is typically largest, followed by the shoulder joint angle, and the wrist joint angle is smallest. A smooth angle change curve indicates there is no movement variation during this period, with the joint angle remaining constant and the upper limb acceleration at zero, bearing only gravitational torque. This phenomenon is corroborated by the torque curve diagram.

The data (Table 3, 4) reveals that, during three typical movements, the average values for both the shoulder and elbow joints fall within the comfort range, and the wrist joint's average value is closely aligned with this zone. Notably, the maximum and minimum value for the angles of the shoulder, elbow, and wrist joints all fall within the human body's range of motion limits. Similarly, the data (Table 5, 6) reveals that, the force comfort index of shoulder and elbow is also above the "more comfort" level, which indicates that the operation comfort of the experimental test meets the design requirements of the core rig.

This experiment employs a virtual driller's cabin model designed using Jack version  8.0.1 software and performs an ergonomic evaluation with Catia software. The study utilizes motion capture sensors to create a digital human, enabling the quantitative analysis of upper limb comfort, visual field, and reachable range, thereby affirming the validity of the previous cabin design. " (Please see page 15, line 350 for the original article modification)

Comments 5: The use of software tools like Unity3D is appropriate, yet the manuscript could elaborate more on the challenges and limitations encountered during implementation.

Response 5: When using Unity3D tool, I did encounter many challenges and limitations, especially when integrating real-time motion capture data in Unity3D, it was necessary to pay attention to hardware performance, data processing load and software scalability, optimize the process such as adopting multi-threading technology, and consider additional plug-ins or custom development to meet complex requirements, so as to ensure data accuracy and real-time performance. In the process of construction, resources were added to the scene by adding the plane model and lights provided by Unity3D, and position, rotation and scale data were adjusted in the transform window. If the spatial layout of the entire scene is not considered in the above operations, it may lead to overlap between resources, affecting the visual effect and interaction design. At the same time, there is also a parent-child relationship between resources, in order to prevent layout confusion, it is necessary to frequently test and preview the effect during the adjustment process to ensure that it meets the design requirements. (Please see page 9, line 230 for the original article modification)

Comments 6: The conclusion suggests future research areas, but these are not sufficiently grounded in the findings or limitations of the current study.

Response 6: Thank you for your valuable suggestions. The future research fields proposed in the conclusion have not been discussed in detail according to the research in this paper, so we have revised the conclusion in a wide range. (Please see page 16, line 391 for the original article modification). For the content of this paper, we mainly introduce an innovative method that combines motion capture with real-time data analysis to evaluate human joint Angle and torque. The results show that this method has the potential to enhance ergonomic design and is applicable in dynamic environments. However, we found certain limitations that provide avenues for future research.

First, while the method has shown effectiveness in capturing and analyzing complex human movements, its accuracy can be further improved using more advanced sensor technology. Future research should explore integrating higher-precision sensors or developing more sophisticated algorithms to improve data accuracy, especially in highly dynamic scenarios.

Secondly, the application of this method in a specific control environment is preliminarily verified. Extending the research to a wider range of real-world applications, such as extreme working conditions or different cultural environments, is critical to assessing its universality and adaptability. These studies can provide valuable insights into the robustness and practicality of the approach in different contexts.

Third, this study uses Unity3D as a software tool for data processing and visualization. While this option proved effective, we observed challenges related to software performance and data processing load. Future research will likely focus on optimizing these technologies, possibly incorporating machine learning or AI-driven approaches to improve the automation and scalability of ergonomic analysis.

Finally, while the direct impact of the approach has been analyzed, its long-term effects on ergonomic design and human health remain unexplored. Future research should investigate the sustainability of this approach applied over time, assessing how continued use may affect ergonomic outcomes and contribute to overall health and well-being.

While this study provides a solid foundation for advancing ergonomic evaluation through real-time data integration, further research is necessary to address its limitations and fully realize its potential in a broader and more complex setting.

3. Additional clarifications

In addition to the revisions made in response to the reviewer’s comments, we have also conducted a thorough review of the manuscript to ensure clarity and coherence throughout the document. We have refined several sections to improve readability. If there are any additional questions or if further clarification is needed, we are more than willing to provide additional information. Thank you once again for your constructive feedback.

Round 2

Reviewer 1 Report

Comments and Suggestions for Authors

The author has fully answered the original questions. The quality of manuscript is substantially improved.